# OpenReview forum: "FlipAttack: Jailbreak LLMs via Flipping"
_ICML.cc/2025/Conference — ICML 2025 poster_

### Official Review · Reviewer_XfGU · 2025-03-05

**Overall Recommendation:** 4

**Summary:**

The authors studied a simple yet effective jailbreak method to attack recent state-of-the-art LLMs in one query. They exploit the auto-regression of LLMs and introduce the left-side perturbation to the text. Four flipping methods are proposed to disguise the harmful content and fool the LLMs. The proposed method is universal, stealthy, and simple. Experiments demonstrate that the proposed method FlipAttack can achieve promising performance.

## update after rebuttal

My concerns are addressed, and I maintain my positive score.

**Claims And Evidence:**

Yes, the claims are supported by experiments and analyses.

**Essential References Not Discussed:**

No.

**Ethics Expertise Needed:**

["Other expertise"]

**Experimental Designs Or Analyses:**

Yes. They design experiments to demonstrate the attack performance, the efficiency of FlipAttack. They analyze the effectiveness of the flipping modes, different parts in FlipAttack. They analyze the deep-in reasons for success of FlipAttack. These designs and analyses are valid.

**Methods And Evaluation Criteria:**

Yes, the proposed methods and evaluation criteria are reasonable.

**Other Comments Or Suggestions:**

1. In Line 16, "that they struggle to comprehend the text when the perturbation is added to the left side" -> "that they struggle to comprehend text when perturbations are added to the left side"
2. In Line 78, "the success rate of 94.04% on GPT-4 Turbo and 86.73% on GPT-4" should clarify that "the success rates are 94.04% for GPT-4 Turbo and 86.73% for GPT-4."

**Other Strengths And Weaknesses:**

**Strengths**
- The motivation (attacking LLMs via 1 query) is clear, and the methodological designs (starting from the property of autoregression) are reasonable.
- The core idea of adding left-side perturbation is novel. Although the method is straightforward, the analyses of designs are sufficient, and the performance is promising.
- The experiments are very comprehensive, demonstrating the performance, efficiency of the proposed method. The reason analyses in section 4.3 and the case studies in the appendix are interesting.
- The paper survey is comprehensive in the related work part.


**Weaknesses**
- Analyzing Figure 4 reveals inconsistent conclusions. For example, CoT helps to improve the attack success performance in Figure (a) (f), but in Figure (e), CoT leads to a significant drop in performance.
- The authors merely conduct experiments on API-based LLMs but overlook the open-sourced models.
- The proposed defense strategies are straightforward but seem not very effective, as shown in Table 21. How to defend the proposed method effectively?
- Is the proposed method a cipher-based jailbreak method? If not, what’s the main difference between the cipher-based jailbreak methods.

**Questions For Authors:**

1. What’s the strength of the proposed method compared to the cipher-based jailbreak attacks? I think the text flipping may be one kind of ciphers.
2. How to defend the proposed attack effectively except the two naive defend methods mentioned in the original paper?

**Relation To Broader Scientific Literature:**

They propose a simple yet effective method to jailbreak LLMs via 1 query. It is universal, stealthy and simple. They analyze the reasons for the success and provide the insights for defense.

**Theoretical Claims:**

They don’t provide the theoretical claims.

---

> ### Author Rebuttal · Authors · 2025-03-30
>
> Thanks for your insightful and constructive review. We response to each question as follows. Following your suggestion, **all modifications will be added to the final paper.**
>
>
> **Inconsistent Conclusions**
> - For different LLMs, **their abilities are different**, which may lead to different conclusions.
> - For example, like GPT 3.5 Turbo, CoT **helps it to better understand** the instruction and recover the flipped text. Meanwhile, CoT does not make it recognize the harmful intent in the prompt.
> - Besides, for GPT-4o mini, CoT may help it to recognize the harmful intent in the prompt, thus decreasing the attack success rate.
>
>
>
> **Open-sourced Models**
> - We conducted experiments on open-sourced models like **LLaMA 3.1 405B** and **Mixtral 8x22B** in the original paper.
>
>
> **How to Defend**
> - We can utilize a **reasoning-base guard model** to defend such flipping-based attacks.
> - We can further enhance the **text understanding ability** of the LLM itself and help it to **recognize the harmful intent** in the attack.
> - We use some **non-autoregressive methods** like your mentioned LLaMA to alleviate the autoregressive nature.
>
> **Cipher-based Method**
> - Our proposed method is not based on the cipher. Our method is based on the analyses of the **autoregressive nature** of the LLMs.
> - The existing cipher-based methods are **typically complex and limit their efficiency and effectiveness**. Differently, our method is simple yet effective.
> - Unlike ciphers, which focus on **hiding content via transformations**, text flipping in the proposed method likely affects the semantic and structural aspects of the input rather than just encoding.
> - Cipher attacks rely on encoding that can be **decoded back into a harmful prompt**. The flipping strategy may instead exploit how LLMs interpret and prioritize different parts of the input.
>
>
>
> **Minors**
> - We will **fix these typos** in the final version.

---

> > ### Comment · Reviewer_XfGU · 2025-04-05
> >
> > I have read the authors’ rebuttal, and my problem has been solved. I decide to keep my score.

---

> > > ### Author Response · Authors · 2025-04-08
> > >
> > > Thanks for your support. We will further improve the quality of our paper according to your suggestions in the final version.

---

### Official Review · Reviewer_L6kD · 2025-03-11

**Overall Recommendation:** 4

**Summary:**

The paper introduces FlipAttack, a novel jailbreak attack method designed for black-box large language models (LLMs). The authors analyze the autoregressive nature of LLMs, revealing that they struggle to comprehend text when perturbations are placed on the left side. Based on this insight, they propose a method to disguise harmful prompts by applying left-side perturbations, leading to four distinct flipping modes. The effectiveness of FlipAttack is demonstrated through extensive experiments on eight different LLMs. The paper claims that FlipAttack is universal, stealthy, and can jailbreak LLMs with a single query.

**Claims And Evidence:**

Yes, they provide clear and convincing evidence for the claims.

**Essential References Not Discussed:**

NA

**Experimental Designs Or Analyses:**

Yes. The designed experiments are valid. Table 1, 2 demonstrates the superiority of the proposed method. Figure 3 shows efficiency. Figure 4 shows the effectiveness of different modules. Table 3,4,5 analyze the underlying mechanisms of FlipAttack.

**Methods And Evaluation Criteria:**

Yes, they make sense.

**Other Comments Or Suggestions:**

1. Keep the overall pages of the whole paper and improve the readability.
2. In section 4.3, the random words like ``Q@+?2gn’’ limit the readability and reduce quality of the paper. Maybe you should move these detailed cases to the appendix.
3. Figure 1 is too small.

**Other Strengths And Weaknesses:**

Pros:
1. The proposed method is simple yet effective, requiring merely one query for jailbreaking LLMs, while most existing methods require multiple queries or an ensemble.
2. The concept of using left-side perturbations to exploit the autoregressive nature of LLMs is quite interesting, which may make this attack hard to defend.
3. The experiments are solid, with significant performance improvement and detailed analysis. The explainable reasons for the successful attack improve the reliability of attacks and provide potential ideas for defending such attacks.
4. The supplementary material is sufficient, and the codes are open-sourced.

Cons:
1. The appendix is too long and contains many case studies, which are not very necessary and limit the readability. Besides, the cases have already been provided in the given code repository.
2. Missing theoretical analyses of the proposed method. Although for such a practical jailbreak, theoretical analysis may be difficult to propose, coming up with theoretical principles for attacks will significantly improve the quality of the paper.
3. On LLaMA 3.1 405B, the attack success rate is limited. While other baselines are not effective, authors should conduct more analyses and provide explanation.
4. Missing citation on recent awesome defense method against jailbreak. [1] I’m curious can the proposed method break this interesting constitutional classifier?
[1] Constitutional Classifiers: Defending against Universal Jailbreaks across Thousands of Hours of Red Teaming

**Questions For Authors:**

1. I’m curious how to defend such flipping attack successfully. Because it is from the nature of the LLMs, autoregression, all of the current sota LLMs are autoregressive, how to defend it from the root.
2. Can the proposed method attack some recent diffusion-based LLMs like LLaDA [1]
[1] Large Language Diffusion Models

**Relation To Broader Scientific Literature:**

Safe of large language models. This paper provides an effective and efficient method to jailbreak LLMs and provides the underlying understanding of the attack.

**Theoretical Claims:**

This paper has not theoretical claims.

---

> ### Author Rebuttal · Authors · 2025-03-30
>
> Thanks for your insightful and constructive review. We response to each question as follows. Following your suggestion, **all modifications will be added to the final paper.**
>
> **Theoretical Analyses**
> - Jailbreak attack is a **practical direction**.
> - We provide **empirical analyses** to demonstrate the **universality** and **stealthy** of the proposed method.
> - We will conduct **theoretical analyses on the autoregressive nature** of the method in the future.
>
>
> **LLaMA 3.1 405B**
> - It **instruction following ability** is limited, and can not recover the flipped sentence and execute the harmful intent.
> - The **safety alignment** of this mode is better than close-sourced models.
> - Other baselines also achieve **unpromising performance** on this model.
>
>
> **Constitutional Classifiers**
> - Please note that this paper is online on **31 Jan 2025**, while the deadline of ICML 2025 is on **09 Jan 2025**. We regard it as **a concurrent work**.
> - We will **discuss constitutional classifiers** in detail in the final version.
> - We will test our method on it **once constitutional classifiers are open-sourced or the API is available**.
>
>
> **How to Defend**
> - We can utilize a **reasoning-base guard model** to defend such flipping-based attack.
> - We can further enhance the **text understanding ability** of the LLM itself and help it to **recognize the harmful intent** in the attack.
> - We use some **non-autoregressive methods** like your mentioned LLaMA to alleviate the autoregressive nature.
>
>
>
> **LLaDA**
> - Please note that this paper is online on **14Feb 2025**, while the deadline of ICML 2025 is on **09 Jan 2025**. We regard it as **a concurrent work**.
> - We will **discuss these kinds of non-autoregressive models** in detail in the final version.
>
>
> **Minors**
> - We will **move the case studies into the code** in the final version.
> - We will **make Figure 1 larger** in the final version.
> - The random words are generated randomly in our experiment. We will **make it more readable** in the final version.

---

> > ### Comment · Reviewer_L6kD · 2025-04-05
> >
> > Thanks. I notice the constitutional classifier and LLaDA are indeed concurrent work. Thus, there is no need to compare them in performance. The provided defense strategies are interesting.
> >
> > As the authors addressed my concerns, I would consider raising my score based on the following reasons. 1) The method is stealthy and universal, achieving a promising attack success rate within 1 query. 2) The empirical analyses of left-side perturbations help authors better understand the mechanism of jailbreaking. 3) The experiments are solid and the codes are available.

---

> > > ### Author Response · Authors · 2025-04-05
> > >
> > > Thanks for your support. We will further improve the quality of our paper according to your suggestions in the final version.

---

### Official Review · Reviewer_KQwB · 2025-03-13

**Overall Recommendation:** 2

**Summary:**

The authors propose FlipAttack, which encodes malicious prompts by reordering words or characters and relies on the reasoning capabilities of the LLM s.t. it can decipher the prompt. The authors demonstrate empirically that this procedure often bypasses the guardrails, is very efficient, and is difficult to detect.

**Claims And Evidence:**

The left-side perturbation claim is somewhat unclear. First, the perplexity experiment (Table 5) is at most suggesting something along these lines. Second, the perturbations are not really happening on the left side of the prompt; rather, the malicious request is reordered in its entirety. Solely due to the "flipping guidance" one might argue that the attack happens early in the prompt.

Other than that, the attack achieves convincing attack success rates, demonstrating that the chosen perturbations and prompt templates are reasonable.

However, some claims like "FlipAttack jailbreaks LLMs with merely 1 query" are unclear since the details of FlipAttack remain vague.

**Essential References Not Discussed:**

E.g., "left side attacks" have also been studied by COLD [A] and PGD [B]. [A] also studies more flexible rewrites.

[A] Guo et al. "COLD-Attack: Jailbreaking LLMs with Stealthiness and Controllability" 2024

[B] Geisler et al. "Attacking Large Language Models with Projected Gradient Descent" 2024

**Experimental Designs Or Analyses:**

It is unclear how the FlipAttack works specifically. The main details of the experimental setup should be in the main body of the work. The authors defer all details to the appendix.

**Methods And Evaluation Criteria:**

Yes.

**Other Comments Or Suggestions:**

1. The set notation for harmful request $\mathcal{X} = \{x_1,x_2,\dots\}$ is odd as it implies that tokens would be unordered.
1. Table 1: "white-box attack" is misleading as these are transfer attacks from a different victim model
1. Line 184. It should probably be "keeping **it** universal"
1. The header contains "Submission and Formatting Instructions for ICML 2025" instead of the paper title
1. Figure 1: text too small
1. Abstract: terms like "flipping modes" are unclear

**Other Strengths And Weaknesses:**

Further weaknesses:
1. The paper remains vague on the details of FlipAttack. The authors propose four rewrites ("attack disguise") and four different templates ("flipping guidance"). In the actual attacks, do the authors try all 16 combinations? How do the authors come up with claims like "FlipAttack jailbreaks LLMs with merely 1 query"? It would be good to provide, e.g., pseudo-code.
1. Essential setup details are missing in the main body
1. It is unclear how inserting gibberish early in a prompt vs. late in the prompt relates to a conclusion w.r.t. left to right understanding.
1. The use of "stealthy" is quite distracting and not really explained until the experimental section.
1. The use of "universality" is unclear.

**Questions For Authors:**

See above as well.

**Relation To Broader Scientific Literature:**

The authors propose a simple and efficient encoding of malicious requests via reordering words and characters. Yet, FlipAttack's pertiurbations are very effective.

**Theoretical Claims:**

Not applicable.

---

> ### Author Rebuttal · Authors · 2025-03-29
>
> Thanks for your insightful and constructive review. We response to each question as follows. Following your suggestion, **all modifications will be added to the final paper.**
>
> **Left-side Perturbation**
> - Left-side perturbation is the **principle idea** of our proposed FlipAttack. It merely helps the readers **better understand how FlipAttack works** and motivates researchers to defense against such an attack.
> - Concretely, our flipping modes can **be regarded as iteratively adding left-side perburbation** to the original prompt, disguising the harmful intents.
> - To help you understand this process, we created a GIF to **figure out the workflow** of our method in https://anonymous.4open.science/r/ICML25-7156-FlipAttack-78D5/flipattack_overview.gif.
> - For the experiments regarding perturbations that really happen on the left side of the prompt, please see **Figure 7**.
> - **'Flipping guidance' is used to help the LLM flip back the flipped prompt and execute the harmful behaviors**. We will revise to **clarify the role of 'flipping guidance'** and alleviate misguiding readers.
>
> **Jailbreak LLM with merely 1 query**
> - In Line 106 of the original paper, we mentioned that `They utilize iterative refinement and cost a large number of queries.`
> - This means the promising performance of the **existing jailbreak attacks relies on multiple queries** to victim LLMs and iterative refinement, e.g., RENELLM, GPTFUZZER, and TAP, leading to **high costs**. Also, it **limits their applicability** since the multiple abnormal queries will decrease the stealthiness of the attack.
> - Differently, our method **can jailbreak LLMs** merely **using 1 query to the victim LLM**, improving the **efficiency and applicability**.
>
> **Details of FlipAttack**
>
> - For the **mode selection**, we have already listed them in **Lines 743-748 of the original paper**.
> - Besides, we have already conducted **ablation studies** on them. See **Figure 4 and Figure 5 of the original paper**.
> - In the actual attack, for the tested LLMs in this paper, the attackers can just **adopt our combination settings**. For a new LLM out of this paper, the attackers can conduct ablation studies on the combinations or just use a default combination to attack the LLM.
> - We will **move more details into the main body** in the final version.
> - We already provide the **excusable code** in https://anonymous.4open.science/r/ICML25-7156-FlipAttack-78D5/README.md
>
> **Related Work**
> - For COLD-Attack, we have already **discussed and compared it in the original paper**. Please check Lines 79, 283, and 420. Besides, **COLD-Attack is a white-box attack** method while our proposed method is a black-box attack method.
> - And **[B] is also a white-box attack**. As they claimed in their paper: `Additionally, we did not conduct experiments against AI assistants deployed for public use, like ChatGPT, Claude, or Gemini. Nor is our attack directly applicable to such models due to the white-box assumption`. We will add a discussion of this paper in the white-box attack part.
>
> **Relationship Between Left-side Perturbation and Left to Right Understanding**
> - First, we invest that the LLMs tend to understand the text from left to right.
> - Then, based on this finding, we aim to destroy the understanding ability of LLMs on the harmful prompts.
> - One effective way is to add noisy text on the left side of the harmful prompt. Adding noise in the right side will influent less.
> - We use the harmful prompt itself to construct this noise via flipping.
> - The whole workflow can be found in **Left-side Perturbation**.
>
> **Stealthy**
> - It means the **attack is hard to be detected** by victim LLM itself or the guard model.
> - It is a **common term for jailbreak attacks** [1,2].
> - We will **add more explanation of stealthy in the early part** of the final paper.
>
>       [1] Stealthy Jailbreak Attacks on Large Language Models via Benign Data Mirroring
>       [2] Gpt-4 is too smart to be safe: Stealthy chat with llms via cipher
>
> **Universality**
> - As we mentioned in Line 47 of the original paper, `First, to make our proposed method universally applicable to state-of-the-art LLMs, we study their common nature, i.e., autoregressive, `.
> - Universality means the proposed **attack method can be effective for all LLMs**.
> - Since **our design is from the autoregressive nature of the LLMs**, they **are all vulnerable to our FlipAttack**.
> - We will **add more explanation of universality in the early part** of the final paper.
>
> **Minors**
> - We will modify the notation. `Given a harmful request X = (x₁, x₂, …, xₙ) as an ordered sequence of n tokens.`
> - In Table 1, we will explain `white-box attack` as transfer attacks in the title.
> - We will fix Line 184 and modify it as `keeping it universal`.
> - We will remove Submission and Formatting Instructions for ICML 2025 and use our title.
> - We will make the text in Figure larger.
> - We will give a detailed definition of flipping modes in the abstract.

---

### Decision · Program_Chairs · 2025-05-01

**Decision:**

Accept (poster)

**Comment:**

The paper concerns jailbreaking attacks in LLMs and presents FlipAttack, a method to disguise harmful prompts using left-side perturbations, making it difficult for LLMs to detect. The intuition is based on the reading order we use of humans. The paper proposes four flipping modes to guide LLMs in executing harmful behaviors. Experiments show FlipAttack's high success and bypass rates across eight LLMs and guard models. The attack is based on a simple heuristic, while the paper is empirical. However, the reviewers agree that this is an effective attack and has not appeared before. Even though I do recommend acceptance, I strongly recommend the authors to consider the comments of the reviewers and the Reviewer KQwB in particular, on making the experiments closer to the standard experiments used in the literature and improving the readability.